# A mutant *Escherichia coli* that attaches peptidoglycan to lipopolysaccharide and displays cell wall on its surface

**Marcin Grabowicz[1], Dorothee Andres[2], Matthew D Lebar[2], Goran Malojčić[2†], Daniel Kahne[2,3]\*, Thomas J Silhavy[1]\***

[1]Department of Molecular Biology, Princeton University, Princeton, United States; [2]Department of Chemistry and Chemical Biology, Harvard University, Cambridge, United States; [3]Department of Biological Chemistry and Molecular Pharmacology, Harvard Medical School, Boston, United States

**Abstract** The lipopolysaccharide (LPS) forms the surface-exposed leaflet of the outer membrane (OM) of Gram-negative bacteria, an organelle that shields the underlying peptidoglycan (PG) cell wall. Both LPS and PG are essential cell envelope components that are synthesized independently and assembled by dedicated transenvelope multiprotein complexes. We have identified a point-mutation in the gene for O-antigen ligase (WaaL) in *Escherichia coli* that causes LPS to be modified with PG subunits, intersecting these two pathways. Synthesis of the PG-modified LPS (LPS\*) requires ready access to the small PG precursor pool but does not weaken cell wall integrity, challenging models of precursor sequestration at PG assembly machinery. LPS\* is efficiently transported to the cell surface without impairing OM function. Because LPS\* contains the canonical vancomycin binding site, these surface-exposed molecules confer increased vancomycin-resistance by functioning as molecular decoys that titrate the antibiotic away from its intracellular target. This unexpected LPS glycosylation fuses two potent pathogen-associated molecular patterns (PAMPs).

**\*For correspondence:** kahne@
chemistry.harvard.edu (DK);
tsilhavy@princeton.edu (TJS)

**Present address:** †Syros
Pharmaceuticals Inc.,
Watertown, USA

**Competing interests:** The
authors declare that no
competing interests exist.

**Reviewing editor:** Eduardo A
Groisman, Yale University/HHMI,
United States

## Main text

A peptidoglycan (PG) cell wall is an essential extracytoplasmic feature of most bacteria (*Singer et al., 1989*); this essentiality has made its biogenesis a fruitful target for antibiotics, including vancomycin and penicillin. The cell wall is directly exposed to the extracellular milieu in Gram-positive bacteria, but is shielded in *Escherichia coli* and other Gram-negative species by a highly selective permeability barrier formed by the outer membrane (OM). The OM prevents influx of antibiotics, such as vancomycin, restricting their access to intracellular targets (*Eggert et al., 2001*; *Ruiz et al., 2005*). Lipopolysaccharide (LPS) forms the surface-exposed outer leaflet of the OM and is key to the barrier function (*Osborn et al., 1972*; *Kamio and Nikaido, 1976*; *Nikaido, 2003*). LPS is a glycolipid consisting of a 'lipid A' anchor within the bilayer, and a set of covalently attached distal 'core' saccharides (*Raetz and Whitfield, 2002*). LPS is made at the cytosolic leaflet of the inner membrane (IM) before being flipped to the periplasmic leaflet (*Zhou et al., 1998*). A transenvelope complex of seven lipopolysaccharide transport proteins (LptABCDEFG) delivers LPS from the IM to the OM (*Ruiz et al., 2009*; *Chng et al., 2010a*). A sub-complex of the β-barrel LptD and lipoprotein LptE resides within the OM and accomplishes the final step of inserting LPS into the outer leaflet (*Chng et al., 2010b*).

A recently described *lptE* mutation (*lptE613*) causes defective LPS transport and leads to increased antibiotic sensitivity (*Malojčić et al., 2014*). To better understand the basis of the *lptE613* defect, we isolated spontaneous suppressors that restored antibiotic resistance. One such vancomycin-resistant suppressor mapped to the *waaL* gene, the product of which is an IM glycosyltransferase that attaches

**eLife digest** Tiny Gram-negative bacteria are one of humankind's deadliest foes, causing infections of wounds and the bloodstream that are very hard to treat. Many Gram-negative bacteria are resistant to several common antibiotics, and the few treatments available that can successfully kill the bacteria are often also toxic to the patients. Understanding how these bacteria elude antibiotics could help scientists develop better, less toxic treatments.

Most bacteria are surrounded by a cell wall that helps protect the bacteria and gives them structure. Many broad-spectrum antibiotics, including penicillin and vancomycin, work by interfering with how this protective wall is built from molecules called peptidoglycans. However, Gram-negative bacteria have an outer membrane that prevents many antibiotics from reaching the cell wall, and so the antibiotics are unable to kill the bacteria.

The outer membrane of Gram-negative bacteria is made up of sugars and fatty molecules called lipids. Recently, scientists discovered a mutation that interferes with the movement of the lipid and sugar molecules that make up the outer membrane, which compromises this protective layer and makes the bacteria more susceptible to antibiotics.

To learn more about how this mutation interferes with the defenses of the Gram-negative bacteria *Escherichia coli*, Grabowicz et al. searched for compensating mutations that can counteract it and restore the antibiotic resistance of these mutant bacteria. The search revealed that a mutation in a gene called *waaL* increases *E. coli*'s resistance to vancomycin, but not to other antibiotics. The gene encodes an enzyme, and the mutant form of the enzyme attaches some peptidoglycans to the surface of the outer membrane instead of incorporating them into the cell wall. The stray peptidoglycans on the cell's surface act as decoys, binding to vancomycin and keeping the drug from reaching its true target—the cell wall. The decoy strategy is similar to a mechanism used by Gram-positive bacteria—which lack a protective outer membrane—to resist vancomycin treatment, which also involves creating sites that bind the drug and keep it from its target.

Vancomycin is not currently used clinically to treat *E. coli* or other Gram-negative infections because these bacteria are naturally quite resistant for other reasons. However, Grabowicz et al.'s findings do demonstrate how quickly bacteria can adapt and produce new defenses to antibiotics when old strategies fail.

O-antigen (O-Ag) oligosaccharides to LPS (*Han et al., 2012*; *Ruan et al., 2012*). Indeed, the suppressed strain is certainly no more vancomycin sensitive than is the corresponding wild-type control (*Figure 1A*). However, this suppressor (*waaL15* herein) was not specific for *lptE613* or even for LPS transport defects. The *waaL15* mutation increases vancomycin resistance in strains carrying *bamB* or *bamE* null mutations that disrupt the OM barrier by causing defects in β-barrel protein assembly (*Figure 1B*) (*Ricci and Silhavy, 2012*). Moreover, *waaL15* also increases vancomycin-resistance even in the wild-type strain (*Figure 1A*). The suppressor does not qualitatively improve the OM barrier, since it did not increase resistance against other antibiotics (*Figure 1B*). So, *waaL15* provides a vancomycin-specific resistance mechanism across different strains.

The domesticated *E. coli* K-12 does not produce the normal substrate (O-Ag) of WaaL (*Liu and Reeves, 1994*) and a *waaL* deletion does not suppress vancomycin sensitivity, indicating that *waaL15* is a gain-of-function mutation. Thus, the WaaL15 mutant O-Ag ligase, which contains an F332S substitution, must have an altered activity. Silver-staining of isolated LPS confirmed that WaaL15 modifies LPS with additional sugars to produce an additional glycoform (LPS*), detected as a higher molecular-weight band that is absent in *waaL*$^+$ (*Figure 2A*). WaaL can use two minor saccharide substrates to modify LPS in *E. coli* K-12: enterobacterial common-antigen (ECA) and colanic acid (CA). ECA-modified LPS is a minor constituent of the OM (*Schmidt et al., 1976*; *Meredith et al., 2007*). Production of CA is regulated by the Rcs phospho-relay stress response system, and CA-modified LPS (called 'M-LPS') is only detectable during severe envelope stress (*Meredith et al., 2007*). Perhaps *waaL15* had improved the utility of one, or both, of these substrates. However, LPS silver-staining revealed that LPS* remained detectable when we inactivated biosynthesis of ECA (*rff*), CA (*cpsG*), or both these polysaccharides (*rff cpsG*) (*Figure 2A*). Moreover, if we increase the amounts of a competing substrate by introducing the *rcsC137* mutation to activate expression of the genes for CA biosynthesis

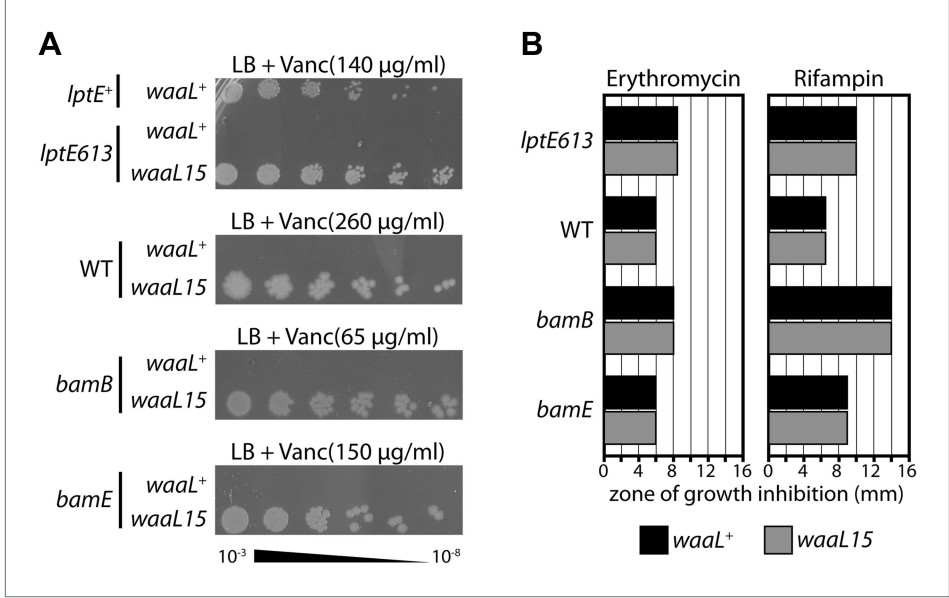

**Figure 1**. A mutant O-antigen ligase increases vancomycin resistance. (**A**) *waaL15* provides a strain-independent increase in vancomycin resistance. Isogenic strains, differing by a point mutation in *waaL*, were plated by serial dilution on LB agar containing indicated amounts of vancomycin. (**B**) *waaL15* does not improve resistance against other antibiotics. Antibiotic discs containing either 15 μg erythromycin or 5 μg rifampin were placed on LB agar overlays inoculated with the indicated strains. Diametric zones of growth inhibition were measured across the disc. The disc diameter was 6 mm and this value represents growth at the disc.

(*Gottesman et al., 1985*), we actually observed lowered LPS* abundance at the expense of increased M-LPS (*Figure 2B*). Notably, the decrease in LPS* correlated with a significant reduction in vancomycin-resistance, providing evidence that LPS* molecules directly mediate the resistance (*Figure 2C*). Similarly, if O-antigen biosynthesis is restored by introducing a wild-type *wbbL* gene, we observe lowered LPS* at the expense of wild-type LPS and vancomycin resistance is reduced. We conclude that WaaL15 is able to use a new substrate and thereby generate a previously uncharacterized LPS glycoform that provides a specific mechanism for vancomycin resistance.

All native WaaL substrates contain carbohydrates linked to a common undecaprenyl (Und) lipid carrier. PG biosynthesis involves a disaccharide pentapeptide (DPP) linked to the same Und carrier, a molecule called lipid II (*Figure 3A*). To directly determine if lipid II is a substrate for WaaL15, we treated isolated LPS* with the muralytic enzyme mutanolysin (*Figure 3A,B*). Digestion of purified LPS*, but not LPS, liberated near-stoichiometric quantities of fragments that were identified by mass spectrometry as DPP or derivatives with a tetrapeptide stem (*Figure 3C*). Importantly, there is no evidence for cross-linked products suggesting that lipid II was the source of the LPS* glycosylation (*Lebar et al., 2013*).

There are several carboxypeptidases in the periplasm that remove the terminal D-Alanine (D-Ala) from DPP to produce the tetrapeptide derivative. Indeed, *E. coli* PG contains negligible amounts of pentapeptide stems (*Figure 3—figure supplement 1*). *Figure 3C* shows that about 50% of the LPS* is sequestered before it can be attacked by one of these carboxypeptidases. It seemed likely that sequestration happens because the molecule is transported from the periplasm to the cell surface.

Peptide stems from adjacent peptidoglycan strands in the cell wall are cross-linked via transpeptidation between the penultimate D-Ala on one stem and a *meso*-diaminopimelic acid (*m*-DAP) residue on a nearby stem (*Vollmer et al., 2008*). Extensive cross-linking produces a rigid macromolecular meshwork that is vital to cell wall function. Vancomycin binds and sequesters the terminal D-Ala-D-Ala residues of a pentapeptide stem in order to inhibit transpeptidation (*Perkins, 1969*). Since LPS* was the product of DPP ligation onto LPS, then this modified glycoform should contain vancomycin binding sites. We assessed the ability of purified LPS* to bind vancomycin in vitro. LPS* was immobilized on a carboxymethylated dextran (CM3) chip and we used surface plasmon resonance to monitor

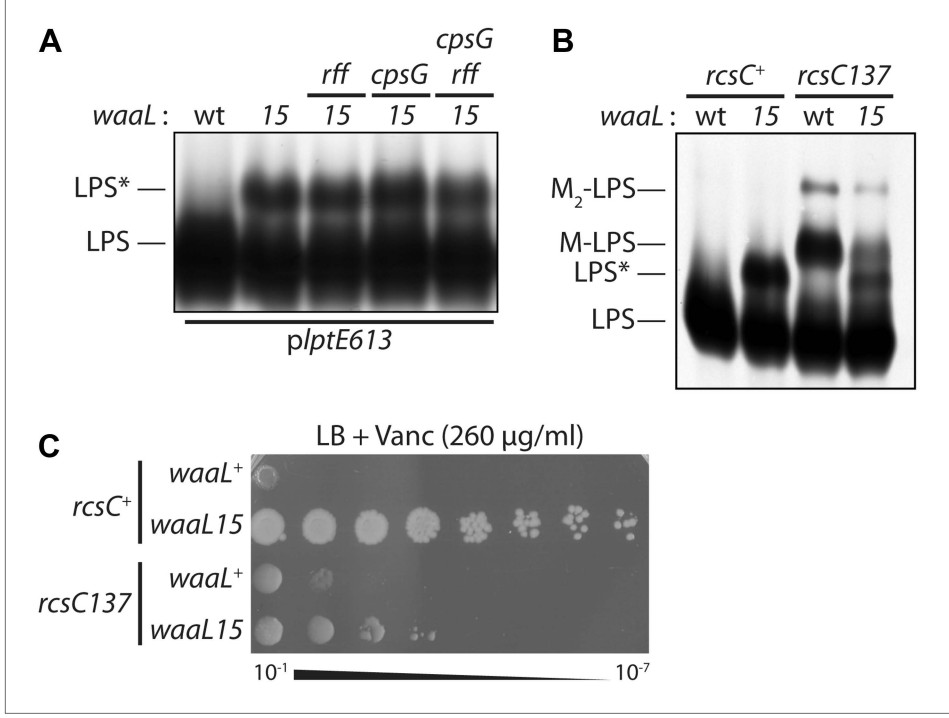

**Figure 2**. Mutant O-antigen ligase produces a novel form of LPS that is directly responsible for vancomycin resistance. (**A**) WaaL15 uses a novel substrate to produce a new LPS glycoform. Isolated LPS was resolved by SDS-PAGE and detected by silver staining. A higher molecular weight glycoform (LPS*) appears in *waaL15* strains. Mutations that inactivate biosynthesis of ECA (*rff*::Tn*10-66*) or CA (Δ*cpsG*::*kan*) do not abrogate LPS* production. (**B**) Overproduction of CA leads to decreased LPS* abundance. Isogenic strains were constructed to express either wt *rcsC*⁺ or the *rcsC137* mutant allele that hyper-activates CA biosynthesis. LPS was isolated and visualized as in (**A**). LPS molecules modified with a one- or two- CA repeat units are labeled M-LPS and M₂-LPS, respectively. (**C**) Reduced LPS* levels correlate with reduced vancomycin resistance. Strains were plated by serial dilution onto LB agar supplemented with vancomycin.

interactions with differing concentrations of vancomycin. We were able to measure specific binding of vancomycin to LPS* and to obtain a $K_d = 0.48 \pm 0.08$ µM (*Figure 4A* and *Figure 4—figure supplement 1*), which is comparable to a reported $K_d$ for vancomycin-lipid II interactions in vesicles (*Al-Kaddah et al., 2010*). Clearly, LPS* molecules include high affinity binding sites for vancomycin.

The ability of LPS* to directly bind vancomycin suggested a possible resistance mechanism, namely that vancomycin is titrated outside the cell. To test this hypothesis, we performed live cell microscopy using a fluorescent vancomycin-BODIPY. We used a wild-type strain background with an intact OM that prevents the influx of vancomycin, to avoid labeling intracellular sites of PG synthesis. Indeed, *waaL*⁺ cells could not be fluorescently labeled (*Figure 4B*). On the other hand, we readily detected circumferential labeling of *waaL15* bacteria, confirming the presence of accessible D-Ala-D-Ala residues at the cell surface (*Figure 4B*).

Several vancomycin-resistance mechanisms exist in Gram-positive bacteria, including: alterations in peptidoglycan metabolism can produce thicker cell walls (*Cui et al., 2003*); and transpeptidation can be reduced to leave more free D-Ala-D-Ala residues within the established cell wall structure (*Sieradzki and Tomasz, 1997*). It was not immediately apparent to us that any of these strategies could be employed in *E. coli* since virtually all of the terminal D-Ala residues of DPP are removed either by cross-linking or by the carboxypeptidases. However, we show that the *waaL15* mechanism is comparable since it also increases the number of free D-Ala-D-Ala targets that can tie up vancomycin. Moreover, by displaying D-Ala-D-Ala at the cell surface the *waaL15* mutation titrates vancomycin away from the true drug target, in an altogether different cellular compartment. Therefore, LPS* confers resistance by acting as a molecular decoy for vancomycin. Given that Gram-negative bacteria are inherently resistant to vancomycin this decoy mechanism may not be of clinical significance. However,

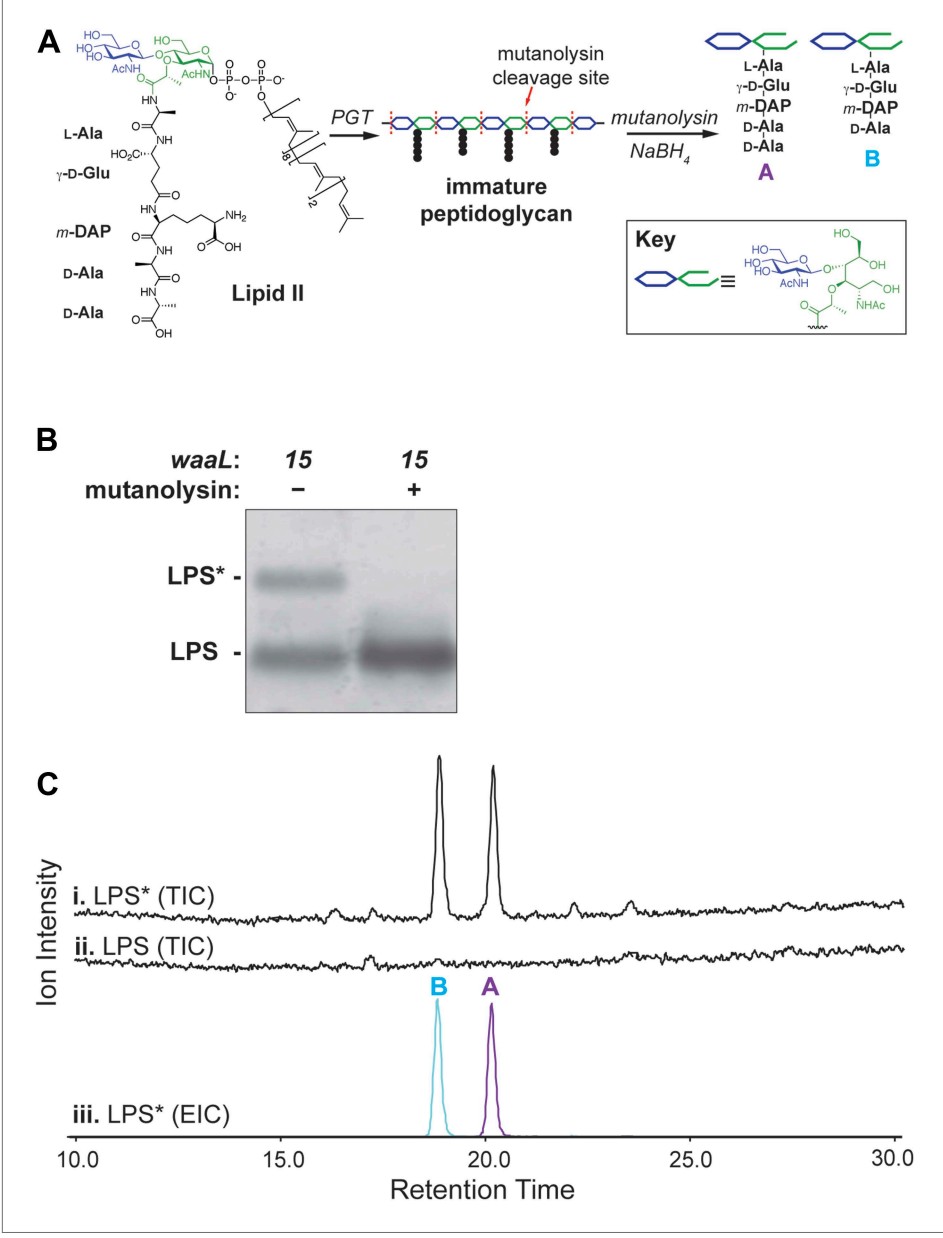

**Figure 3**. Lipid II is the glycosyl donor for LPS*. (**A**) Structure of lipid II and schematic of peptidoglycan cleavage by mutanolysin that releases disaccharide pentapeptide ('A') and tetrapeptide ('B') species. (**B**) Treatment of *waaL15* isolated LPS with mutanolysin cleaves the LPS* modification. (**C**) LPS* is glycosylated with equivalent amounts of lipid II-sourced disaccharide pentapeptide and tetrapeptide. Isolated and purified LPS* from *waaL15* and LPS from *waaL⁺* were treated with mutanolysin and analyzed by LC-MS. Total ion chromatogram for degradation products (i and ii), and the extracted ion chromatogram for LPS* degradation (iii) are shown. M+H and (M+2H)/2 ions corresponding to each fragment were extracted (A: 1013.3 + 507.2; B: 942.3 + 471.7).

The following figure supplement is available for figure 3:

**Figure supplement 1**. The *waaL15* mutation does not affect the PG cell wall.

---

the increased resistance it does confer clearly demonstrates the tremendous adaptability of bacteria under antibiotic stress.

The biosynthesis of LPS* is remarkable. Lipid II in *E. coli* is extremely scarce, its steady-state abundance is thought to be only 1000–2000 molecules per cell (***van Heijenoort et al., 1992***). Insertion

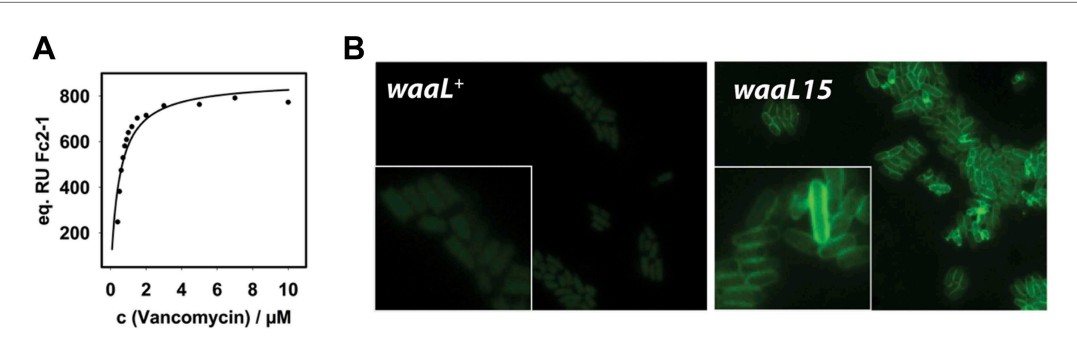

**Figure 4**. Mutant WaaL attaches peptidoglycan fragments to LPS. (**A**) LPS* specifically binds vancomycin. Purified LPS* was immobilized on a CM3 chip and varying concentrations of vancomycin were applied. Binding was measured at 25°C by surface plasmon resonance. Fitting of equilibrium signal yielded a $K_d$ = 0.48 ± 0.08 μM. Standard deviation was measured for 0.6 μM and 1.2 μM and was ±1 RU. (**B**) Vancomycin binds to LPS* at the cell surface. Live exponential-phase growing cells labeled with 1 μg/ml vancomycin-BODIPY for 10 min. Cells were spotted onto M63 minimal medium agar pads and imaged by fluorescence microscopy.
The following figure supplement is available for figure 4:
**Figure supplement 1**. SPR binding kinetics at 25°C.

of new PG is thought to occur via large multiprotein morphogenic complexes: the elongasome and the divisome, responsible for PG synthesis along the lateral cell body and at the septum, respectively. In order to overcome the scarcity of lipid II and limit its diffusion away from sites of PG growth, both complexes are suggested to include at least some of the lipid II biosynthetic enzymes, and the presumed flippases that deliver lipid II from the site of synthesis in the cytoplasm to the site of cell wall assembly in the periplasm (*Szwedziak and Löwe, 2013*). In this model, the substrate for PG synthesis would be isolated physically from the LPS assembly pathway. LPS is inserted into the OM of each cell at a rate exceeding 70,000 molecules per minute (*Lima et al., 2013*) and we approximate that one-third of LPS is modified by WaaL15 with lipid II-sourced DPP. Clearly, WaaL15 has ready access to lipid II and this is inconsistent with a model of diffusion-limited lipid II sequestered at the elongasome or divisome complexes. Recent evidence also points to wider lipid II availability (*Lee et al., 2014*; *Sham et al., 2014*). Our data indicate that the re-charging of the lipid carrier with new DPP must also be extremely efficient to maintain such a robust pool of PG precursor.

WaaL15 drains the available lipid II pool with no apparent detriment to cell wall integrity (*Figure 3—figure supplement 1*). Lipid II limitation can be revealed by synthetic genetic interactions in a strain lacking the elongasome (*Paradis-Bleau et al., 2014*), but it is not the recharging of lipid II that is limiting, rather it is the biosynthesis of DPP (*Supplementary file 1A*).

In many bacteria, LPS is decorated with highly variable O-Ags that are linear polymers of repeating units of 3–6 monosaccharides (*Kalynych et al., 2014*). In *E. coli* the multitude of different O-Ags initiate with GlcNAc, ECA also initiates with GlcNAc. In *E. coli* K-12 when colonic acid is overproduced M-LPS is made from an intitating Glc residue. The F332S mutation broadens substrate specificity of the WaaL glycosyltransferase allowing it to efficiently accept a significantly more bulky initiating MurNAc with an attached oligopeptide stem. The only other glycosyltransferase that is known to use lipid II as a substrate is PglL from *Neisseria* and the use required overproduction of the enzyme in *E. coli* (*Faridmoayer et al., 2008*). It is also remarkable that we detect no OM biogenesis defect in strains carrying *waaL15*, demonstrating that the Lpt system is fully competent for the transport and assembly of LPS* despite the addition of both unnatural sugars and peptide stems. Both LPS and PG are pathogen-associated molecular patterns (PAMPs) that potently activate innate immune responses via distinct pathways, and it seems sensible for Gram-negative bacteria to keep these entities separated. We suggest that the F332S substitution has inactivated an exclusion mechanism that prevents WaaL from utilizing the lipid II pool.

## Materials and methods

### Bacterial strains and growth conditions

Strains and plasmids used in this study are listed in *Supplementary file 1B* and *Supplementary file 1C*, respectively. Strains were grown in lysogeny broth (LB, 10 g/L NaCl) or M63 minimal medium under aeration at 37°C unless otherwise noted. When appropriate, media were supplemented with kanamycin (Kan, 25 µg/ml), ampicillin (Amp, 25–125 µg/ml), tetracycline (Tet, 20 µg/ml), chloramphenicol (Cam, 20 µg/ml), vancomycin (Vanc, 65–260 µg/ml) and arabinose (Ara, 0.2% vol/vol). Kanamycin deletion-insertion mutations of *bamE*, *cpsG*, *mrcA*, *mrcB*, *lpoA* and *lpoB* were obtained from the Keio collection (*Baba et al., 2006*). ECA null *rff*::Tn*10*-66 allele was obtained from strain 21566 (*Meier-Dieter et al., 1990*). The *ompC*::Tn*5 rcsC137* was obtained from strain SG20803 (*Brill et al., 1988*). Mutant alleles were introduced by P1*vir* transduction.

### Isolation and identification of *waaL15*

Spontaneous suppressor mutants of strain MG1029 capable of growing on LB plates supplemented with vancomycin (140 µg/ml) were isolated; one such mutant strain was MG1088. The mutation locus conferring vancomycin-resistance in MG1088 was identified by linkage mapping using a library of random mini-Tn*10* insertions (*Kleckner et al., 1991*). In this way, the *tdh*::Tn*10* allele was found to be approximately 70% linked to the suppressor mutation *waaL15*. The F332S mutation was then identified by PCR amplification and sequencing of the *waaL* locus. The *waaL15* mutation was moved into the NR754 wild-type strain by linkage with *tdh*::Tn*10*. In order to generate the unmarked *waaL15* strain (MG1643) and its wild-type control (MG1642), the *tdh*::Tn*10* mutation was removed from strains MG1210 and MG1211 by first introducing a linked Δ*cysE*::*kan* mutation (*Baba et al., 2006*), selecting for Kan^R and screening for Tet^S transductants that were Vanc^R (*waaL15*) or Vanc^S (*waaL*^+). The Δ*cysE*::*kan* mutation was then replaced with *cysE*^+ by transduction, selection on M63 minimal medium, and screening of Vanc^R/Vanc^S.

### Assessment of genetic linkage by co-transduction

In *E. coli*, two key PG synthases, PBP1A (*mrcA*/*ponA*) and PBP1B (*mrcB*/*ponB*), incorporate DPP from Lipid II into PG strands and also mediate transpeptidation (*Paradis-Bleau et al., 2010*; *Typas et al., 2010*). Recent evidence suggests that lipid II limitation can be revealed by synthetic genetic interactions in a strain lacking *mrcB* (*Paradis-Bleau et al., 2014*). The genetic interaction of PG synthase mutants with *waaL15* was assessed as follows. Kan^R-marked null alleles of *lpoA, lpoB, mrcA* and *mrcB* were introduced by P1*vir* transduction into CAG strains that carry a Tn*10* insertion in a nearby locus (see *Supplementary file 1A*). Kan^R Tet^R transductants were isolated and used to generate P1*vir* lysates. These P1*vir* were used to transduce *waaL*^+ (MG1642) or *waaL15* (MG1643) strains, selecting for the Tn*10* marker. The frequency with which the Kan^R-marked *lpo* and *mrc* alleles were co-transduced (genetically linked) was determined by replica plating on LB+Kan. Linkage was assessed in a total of 300 transductants from three independent experiments. A decrease in the cotransduction frequency in *waaL15* strains relative to *waaL*^+ indicates a synthetic interaction between *waaL15* and the Kan^R-marked allele. The synthetic interaction between *waaL15* and *mrcB*/*lpoB* null alleles was relieved in strains carrying pMurA when expression of the cloned *murA* gene (encoding the enzyme responsible for the first committed step in DPP biosynthesis) was induced with 100 µM isopropyl β-D-1-thiogalactopyranoside (IPTG). Overexpression of *murA* increases the cellular pool of UDP-MurNAc-pentapeptide and consequently also increases the pool of lipid II.

### Analysis of LPS by SDS-PAGE and silver staining

A total to $1 \times 10^9$ cells from liquid culture were taken, pelleted and resuspended 0.05 ml of 'LPS Sample Buffer' (0.66 M Tris pH 7.6, 2% vol/vol sodium dodecyl sulfate [SDS], 10% vol/vol glycerol, 4% vol/vol β-mercaptoethanol, 0.1% wt/vol bromophenol blue). Samples were boiled for 10 min and allowed to cool to room temperature. 10 µl of Proteinase K (2.5 mg/ml, in LPS Sample Buffer) was added and samples were incubated at 56°C for 16 hr. LPS samples were then resolved by SDS-PAGE and silver stained as described previously (*Tsai and Frasch, 1981*). By quantifying band density using ImageJ, we measured that LPS* constitutes 29 ± 1% of the total LPS in *waaL15* samples.

### Antibiotic disc diffusion assay

3 ml of molten LB Top agar (0.75% agar) was inoculated with 0.1 ml of overnight culture. The mixture was poured onto a LB agar plate (1.5% agar,) and allowed to set. Antibiotic discs (BD Sensi-Disc) were

placed on the Top agar overlay and plates were incubated overnight at 37°C. The 'zone of growth inhibition' was measured across the antibiotic disc.

## Fluorescence microscopy

Overnight cultures were sub-cultured at 1:100 into fresh LB broth and grown for 1.5 hr. A 1 ml aliquot was taken, pelleted and was twice washed with 1 ml M63 medium. Cells were resuspended in 0.1 ml of M63 broth containing 1 µg/ml of vancomycin-BODIPY-FL (Life Technologies, V-34850). Cells were incubated at room temperature for 10 min and then washed twice with 1 ml M63 broth. Cells were then resuspended in 0.03 ml of M63 broth, and approximately 2 ml was spotted onto an M63-agarose pad. Cells were immediately visualized on a Nikon Eclipse 90i microscope with a Nikon Plan Apo 1.4/100 × Oil Ph3 phase objective.

## LPS Purification

*E. coli* MG1210 and MG1211 were each grown in 4 × 1.5 l LB medium shaking at 37°C overnight to stationary phase. The cells were harvested by centrifugation for 15 min at 5000×*g*, 4°C and washed with water (700 ml) and ethanol (40 ml) once, then twice with acetone (40 ml). After drying the cell pellet in a desiccator over night in vacuo, PCP (Phenol-Chloroform-petroleum ether) method was used for rough LPS extraction (*Galanos et al., 1969*).

## PG Purification

*E. coli* MG1210 and MG1211 were each grown in 500 ml LB medium shaking at 37°C to stationary phase (6 hr). The cell wall was isolated from the culture as described by *Glauner et al. (1988)* and *Uehara et al. (2009)*, with modifications described below. The cells were resuspended in 20 ml phosphate buffered saline (PB, pH = 7.4) and boiled for 30 min in 80 ml 5% SDS. After the samples cooled, they were pelleted (14,000 rpm, 25°C, 1 hr) and washed six times by pelleting (14,000 rpm, 25°C, 1 hr) from 50 ml water aliquots to remove the SDS. The samples were resuspended in 1 ml PBS, treated with α-amidase (100 µl, 2 mg/ml stock in 50% glycerol, Sigma A-6380) and incubated at 37°C with shaking for 2 hr. To cleave proteins attached to the cell wall, α-chymotrypsin (100 µl, 3 mg/ml in 50% glycerol, Sigma C3142) was added, and the samples were incubated at 37°C with shaking overnight. An additional aliquot of α-chymotrypsin (100 µl) was added, and the samples were digested for an additional 4 hr. To remove the proteins, SDS was added to a final concentration of 1%, and the samples were incubated at 95°C for 1 hr. After cooling, the samples were again pelleted (14,000 rpm, 25°C, 1 hr) and washed with water repeatedly (4 × 25 ml) to remove the SDS. The final peptidoglycan (PG) samples were resuspended in 500 µl 0.02% azide and stored at 4°C.

## Mutanolysin digestion and analysis

The PG composition was analyzed by LC/MS as previously described (*Lebar et al., 2013*). The method was also used to analyze LPS samples. The glycosylhydrolase mutanolysin liberated DPP and disaccharide tetrapeptide from LPS*. Aliquots (40 µl) of PG (from MG1210 and MG1211) and LPS (from MG1210 and MG1211) were incubated with mutanolysin (10 U, 2.5 µl, 4000 U/ml, Sigma M9901, stored at −20°C in 50 mM TES, pH 7.0, 1 mM MgCl$_2$, 10% glycerol) in 50 mM sodium phosphate buffer (pH 6.0, 100 µl total volume) at 37°C with shaking overnight. Another aliquot of mutanolysin (10 U, 2.5 µl) was added, and the mixture was incubated at 37°C with shaking for 3 hr. Insoluble particles were separated by centrifugation (16,000×*g*). The supernatant, containing soluble fragments, was treated with sodium borohydride (10 mg/ml in water, 100 µl) at room temperature for 30 min. Phosphoric acid (20%, 12 µl) was then added to adjust pH to ~4. When bubbling ceased, the samples were lyophilized and re-dissolved in 25 µl water, which was analyzed on LC/MS. LC/MS analysis was conducted with ESI-MS operating in positive mode. The instrument was equipped with a Waters Symmetry Shield RP18 column (5 µm, 3.9 × 150 mm) with matching column guard. The fragments were separated using the following method: 0.5 ml/min H$_2$O (0.1% formic acid) for 5 min followed by a gradient of 0% ACN (0.1% formic acid)/H$_2$O (0.1% formic acid) to 20% ACN (0.1% formic acid)/ H$_2$O (0.1% formic acid) over 40 min.

## Surface plasmon resonance analysis

Purified LPS (0.5 mg/ml) from strains MG1210 or MG1211 were extruded in 20 mM Tris/HCl pH 8, 150 mM NaCl and immobilized on poly-L-lysine coated CM3 Biacore chips on the active and reference channel, respectively (*Malojčić et al., 2014*). All experiments were performed using a Biacore X100

instrument at 25°C at a flow rate of 10 µl/min with 20 mM Tris/HCl pH 8, 150 mM NaCl buffer. Different concentrations of vancomycin were injected for 400 s and dissociation was recorded for another 500 s to return to baseline. No binding was observed to the reference channel. The equilibrium signal in the difference channel was fitted to $f = Bmax*abs(x)/(Kd + abs(x))$ with $R^2 = 0.88$. Standard deviation was measured for 0.6 µM and 1.2 µM vancomycin in triplicate and did not exceed 1 RU.

## Acknowledgements

We thank Natividad Ruiz, Kerrie L May, Robert S Dwyer, and members of the Silhavy lab for insights, comments and suggestions.

## Additional information

### Funding

| Funder | Grant reference number | Author |
|---|---|---|
| National Institutes of Health | GM034821 | Marcin Grabowicz, Thomas J Silhavy |
| National Institutes of Health | GM066174 | Dorothee Andres, Matthew D Lebar, Daniel Kahne |
| Swiss National Science Foundation | P300P2_147905 | Goran Malojčić |
| National Institutes of Health | GM103056 | Matthew D Lebar |
| Swiss National Science Foundation | PA00P3_134194 | Goran Malojčić |

The funders had no role in study design, data collection and interpretation, or the decision to submit the work for publication.

### Author contributions

MG, DA, MDL, Conception and design, Acquisition of data, Analysis and interpretation of data, Drafting or revising the article; GM, Conception and design; DK, TJS, Conception and design, Analysis and interpretation of data, Drafting or revising the article

## Additional files

### Supplementary file

• Supplementary file 1. (**A**) Synthetic interactions between *waaL15* and mutations affecting the elongasome due to limited lipid II availability. (**B**) Strains used in this study. (**C**) Plasmids used in this study.

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
