## [Decision Letter]

Thank you for sending your work entitled “A mutant *E. coli* attaches peptidoglycan to lipopolysaccharide displaying cell wall on its surface” for consideration at *eLife*. Your article has been favorably evaluated by Michael Marletta (Senior editor), a Reviewing editor, and 3 reviewers.

The Reviewing editor and the reviewers discussed their comments before we reached this decision, and the Reviewing editor has assembled the following comments to help you prepare a revised submission.

This manuscript describes an unexpected property of a mutant in the O-antigen ligase *waaL* gene, which specifies a protein that connects the O-antigen with the lipid A core to form the complete LPS molecule. The authors identify an allele of *waaL* that has a different specificity as it can now incorporate a peptidoglycan fragment that can be displayed on the outer membrane of *E. coli*. The strain with the new allele of *waaL* can sequester vancomycin molecules resulting in an increase in resistance to this antibiotic.

Specific comments:

1) The idea that redirecting peptidoglycan to the surface-exposed leaflet of Gram-negative bacteria could be a drug resistance (decoy) mechanism against vancomycin does not seem likely given all Gram-negative bacteria are already incredibly resistant to the drug. Clinically, vancomycin would never be used to treat a Gram-negative bacterial infection. Furthermore, the *E. coli* laboratory strain that this work was performed in has an inactivated O-antigen biosynthetic pathway. Actually, the MIC of their *E. coli* strain for vancomycin is >100 higher than that of vancomycin resistant gram-positive bacterial. Thus, the authors should clarify what the increase in vancomycin resistance means in this particular case.

2) If the *waaL15* allele is introduced into an *E. coli* that makes a complete O antigen, can WaaL still add PG to LPS?

3) Can you please elaborate on the mutanolysin result in the context of the enzyme's specificity? Here, the enzyme is not cleaving within a PG chain but instead at the linkage site between DPP and the LPS acceptor and this is a bit surprising to me (although the data is clear). In K-12, the attachment site for O antigen is thought to be an outer core Heptose residue, although it has not been analyzed in detail. From the specificity of K-12 ligase (Ruan, 2012), mutanolysin must be cleaving a *β*-DPP-1,7-*α*-Hep linkage in the current situation. What is known about mutanolysin substrate specificity?

4) In discussing the flexibility of WaaL (at the end of the Discussion), the authors should comment more directly on the existing specificity of the K-12 ligase. In *E. coli* K-12, the O antigen that K-12 used to have is built on GlcNAc (Stevenson et al., 1994, J Bacteriol, 176: 4144) using the same enzyme that initiates ECA. For M-LPS, its range of activity extends to a Glc initiated polymer (Garegg et al., 1971. Acta Chem Scan, 25: 2103). The shift to recognizing MurNAc-pentapeptide is indeed a significant change.

5) Capture of lipid II by a non-PG glycosylation system has been reported previously ([8], JBC, 283: 34596). There the authors introduced an oligosaccharyltransferase from Neisseria into K-12 and showed it could glycosylate a pilin protein acceptor with PG subunits. This heterologous system is worth mentioning in the Discussion.

---

## [Author Response]

*1) The idea that redirecting peptidoglycan to the surface-exposed leaflet of Gram-negative bacteria could be a drug resistance (decoy) mechanism against vancomycin does not seem likely given all Gram-negative bacteria are already incredibly resistant to the drug. Clinically, vancomycin would never be used to treat a Gram-negative bacterial infection. Furthermore, the* E. coli *laboratory strain that this work was performed in has an inactivated O-antigen biosynthetic pathway. Actually, the MIC of their* E. coli *strain for vancomycin is >100 higher than that of vancomycin resistant gram-positive bacterial. Thus, the authors should clarify what the increase in vancomycin resistance means in this particular case*.

We have been careful to describe the *waaL15* phenotype as “increasing resistance” against vancomycin, taken to mean that the mutation allows strains to grow in the presence of vancomycin concentrations that would kill wild-type *waaL* equivalent strains. We have deliberately not drawn implications of this mutation to clinical resistance. However, to further clarify this issue, we have added the following two sentences: “Given that Gram-negative bacteria are inherently resistant to vancomycin this decoy mechanism may not be of clinical significance. However, the increased resistance it does confer clearly demonstrates the tremendous adaptability of bacteria under antibiotic stress.”

*2) If the* waaL15 *allele is introduced into an* E. coli *that makes a complete O antigen*, *can WaaL still add PG to LPS?*

When we restore O-antigen biosynthesis in our strains (making them *wbbL*^+^), *waaL15* remains able to produce LPS* and improve resistance against vancomycin. In these strains the absolute abundance of LPS* is reduced, presumably because O-antigen is both a highly abundant and preferred substrate. The following sentence clarifying this point has been added: “Similarly, if O-antigen biosynthesis is restored by introducing a wild-type *wbbL* gene, we observe lowered LPS* at the expense of wild-type LPS and vancomycin resistance is reduced.”

*3) Can you please elaborate on the mutanolysin result in the context of the enzyme's specificity? Here, the enzyme is not cleaving within a PG chain but instead at the linkage site between DPP and the LPS acceptor and this is a bit surprising to me (although the data is clear). In K-12, the attachment site for O antigen is thought to be an outer core Heptose residue, although it has not been analyzed in detail. From the specificity of K-12 ligase (Ruan, 2012)*, *mutanolysin must be cleaving a*
*β**-DPP-1,7-**α**-Hep linkage in the current situation. What is known about mutanolysin substrate specificity?*

We agree that the mutanolysin result is surprising. However, since we don’t yet know where DPP is attached to the LPS core, it is difficult for us to comment on the mutanolysin substrate specificity and we’d rather not speculate.

*4) In discussing the flexibility of WaaL (at the end of the Discussion), the authors should comment more directly on the existing specificity of the K-12 ligase. In* E. coli *K-12, the O antigen that K-12 used to have is built on GlcNAc (**Stevenson et al., 1994**, J Bacteriol, 176: 4144) using the same enzyme that initiates ECA. For M-LPS, its range of activity extends to a Glc initiated polymer (**Garegg et al., 1971**. Acta Chem Scan 25: 2103). The shift to recognizing MurNAc-pentapeptide is indeed a significant change*.

We have expanded our discussion of known WaaL substrate specificity by adding the following sentence: “In *E. coli* the multitude of different O-antigens initiate with GlcNAc, ECA also initiates with GlcNAc. In *E. coli* K-12 when colonic acid is overproduced M-LPS is made from an inititating Glc residue.”

*5) Capture of lipid II by a non-PG glycosylation system has been reported previously (*[8]*, JBC, 283: 34596). There the authors introduced an oligosaccharyltransferase from Neisseria into K-12 and showed it could glycosylate a pilin protein acceptor with PG subunits. This heterologous system is worth mentioning in the Discussion*.

We have amended the text to include this reference by adding this sentence: “The only other glycosyltransferase that is known to use lipid II as a substrate is PglL from Neisseria and the use required overproduction of the enzyme in *E. coli* (8).”